# Adipokinetic hormone signaling mediates the enhanced fecundity of *Diaphorina citri* infected by '*Candidatus* Liberibacter asiaticus'

**Jiayun Li[1], Paul Holford[2], George Andrew Charles Beattie[2], Shujie Wu[1], Jielan He[1], Shijian Tan[1], Desen Wang[1], Yurong He[1], Yijing Cen[1]\*, Xiaoge Nian[1,3]\***

[1]National Key Laboratory of Green Pesticide, Department of Entomology, College of Plant Protection, South China Agricultural University, Guangzhou, China; [2]School of Science, Western Sydney University, Penrith, Australia; [3]Henry Fok School of Biology and Agriculture, Shaoguan University, Shaoguan, China

**\*For correspondence:**
cenyj@scau.edu.cn (YC);
nianxiaoge0629@126.com (XN)

**Competing interest:** The authors declare that no competing interests exist.

**Abstract** *Diaphorina citri* serves as the primary vector for '*Candidatus* Liberibacter asiaticus (*C*Las),' the bacterium associated with the severe Asian form of huanglongbing. *C*Las-positive *D. citri* are more fecund than their *C*Las-negative counterparts and require extra energy expenditure. Therefore, understanding the molecular mechanisms linking metabolism and reproduction is of particular importance. In this study, we found adipokinetic hormone (*DcAKH*) and its receptor (*DcAKHR*) were essential for increasing lipid metabolism and fecundity in response to *C*Las infection in *D. citri*. Knockdown of *DcAKH* and *DcAKHR* not only resulted in the accumulation of triacylglycerol and a decline of glycogen, but also significantly decreased fecundity and *C*Las titer in ovaries. Combined in vivo and in vitro experiments showed that miR-34 suppresses *DcAKHR* expression by binding to its 3' untranslated region, whilst overexpression of miR-34 resulted in a decline of *DcAKHR* expression and *C*Las titer in ovaries and caused defects that mimicked *DcAKHR* knockdown phenotypes. Additionally, knockdown of *DcAKH* and *DcAKHR* significantly reduced juvenile hormone (JH) titer and JH signaling pathway genes in fat bodies and ovaries, including the JH receptor, *methoprene-tolerant* (*DcMet*), and the transcription factor, *Krüppel homolog 1* (*DcKr-h1*), that acts downstream of it, as well as the egg development related genes *vitellogenin 1-like* (*DcVg-1-like*), *vitellogenin A1-like* (*DcVg-A1-like*) and the vitellogenin receptor (*DcVgR*). As a result, *C*Las hijacks AKH/AKHR-miR-34-JH signaling to improve *D. citri* lipid metabolism and fecundity, while simultaneously increasing the replication of *C*Las, suggesting a mutualistic interaction between *C*Las and *D. citri* ovaries.

## eLife assessment

This **important** study reveals the molecular basis of mutualism between a vector insect and a bacterium responsible for the most devastating disease in citrus agriculture worldwide. The evidence supporting the conclusions is **compelling**, with **solid** biochemical and gene expression analyses demonstrating the phenomenon. We believe this work will be of great interest to the fields of vector-borne disease control and host-pathogen interaction.

## Introduction

The majority of devastating plant pathogens, including viruses and bacteria, are transmitted by insect vectors (*Eigenbrode et al., 2018*). In recent decades, the interactions between vectors and pathogens have been widely studied (*Eigenbrode et al., 2018*; *Mao et al., 2019*; *Berasategui et al., 2022*; *Ray and Casteel, 2022*). However, studies on the mechanisms underlying these interactions have mainly focused on vector-virus or vector-fungus interactions; less research has been performed on vector-bacteria interactions. Knowledge of the relationships between vectors and pathogens is of great significance for understanding the epidemiology of plant pathogens, and this knowledge may help develop new strategies for controlling both vectors and pathogens. The severe Asian form of huanglongbing (HLB) is currently the most destructive citrus disease in Asia and the Americas where it is associated with the phloem-restricted, Gram-negative, α-Proteobacteria *C*Las and transmitted by the Asian citrus psyllid, *Diaphorina citri* Kuwayama (Hemiptera: Psyllidae) (*Capoor and Rao, 1967*; *Gottwald, 2010*; *Bové, 2014*; *da Graça et al., 2016*; *Fuentes et al., 2018*). However, since *C*Las has not been cultured in vitro, Koch's postulates have not been proven. Control of *D. citri* is the most effective way to prevent HLB epidemics (*Gottwald, 2010*; *Bové, 2014*) but experience in Asia (*Beattie, 2020*) and the United States of America (*Graham et al., 2020*; *Li et al., 2020*), and recent press reports from Brazil, indicate that it is impossible to suppress psyllid populations with insecticides to levels that prevent spread of *C*Las.

Previous studies have reported that infection with *C*Las significantly increases the fecundity of *D. citri* (*Pelz-Stelinski and Killiny, 2016*; *Ren et al., 2016*; *Wu et al., 2018*). Reproduction is a costly process in terms of energy usage in the adult life of female insects (*Arrese and Soulages, 2010*), and there is an inevitable trade-off between lipid storage and use, because animals mobilize their lipid reserves during reproduction. However, the mechanism of how *D. citri* maintains a balance between lipid metabolism and increased fecundity after infection with *C*Las remains unknown. In insects, energy mobilization is under control of adipokinetic hormone (AKH), which was the first insect neurohormone to be identified; it was shown to stimulate lipid mobilization for locomotor activity in locusts (*Stone et al., 1976*). To date, more than 80 different forms of AKH have been identified or predicted in arthropod species due to similar structural characteristics of the enzyme (*Toprak, 2020*). The mechanisms of synthesis and biological activity of AKH have also been investigated. Insect store lipids in the form of triacylglycerides (TAG) and as carbohydrates in the form of glycogen in their fat bodies (*Azeez et al., 2014*). Under conditions of energy demand, AKHs are synthesized in the corpora cardiaca, released into the hemolymph, and bind to receptors (AKHRs) located on the plasma membrane of fat body cells. AKHR binding and activation triggers the release of energy-rich substrates after which various cellular signaling pathway components come into play (*Gáliková et al., 2015*). AKHRs are a class of G protein-coupled receptors and were first identified in *Drosophila melanogaster* (*Park et al., 2002*) and *Bombyx mori Staubli et al., 2002*; subsequently, they were identified in a large number of insect species (*Ziegler et al., 2011*; *Attardo et al., 2012*; *Konuma et al., 2012*; *Jedlička et al., 2016*).

While the AKH pathway is recognized for its role in mobilizing stored lipids, its involvement in vector-pathogen interactions remains understudied. For example, in *Anopheles gambiae-Plasmodium falciparum* interactions. *P. falciparum* infection influences *A. gambiae* AKH signaling, lipid metabolism, and mobilization of energy reserves to support the parasite's metabolic and structural needs (*Nyasembe et al., 2023*). In another example, infection of cockroaches with the fungus *Isaria fumosorosea* result in a significant rise in AKH levels in the central nervous system in response to stress responses (*Karbusová et al., 2019*). The functions of the AKH/AKHR in insect-pathogen interactions need further investigation. Recent studies have shown that AKH/AKHR signaling plays a regulatory role during female reproduction in insects. For example, depletion of AKHR in *Bactrocera dorsalis* resulted in TAG accumulation, decreased sexual courtship activity, and fecundity (*Hou et al., 2017*). AKHR knockdown in *Nilaparvata lugens* interferes with trehalose homeostasis and vitellogenin uptake by oocytes, which causes delayed oocyte development and reduced fecundity (*Lu et al., 2018*). AKH/AKHR signaling also regulates vitellogenesis and egg development in *Locusta migratoria* via triacylglycerol mobilization and trehalose homeostasis (*Zheng et al., 2020*). As outlined above, the fecundity of *D. citri* is increased by *C*Las. Whether AKH/AKHR signaling participates in this increase in fecundity due to infection via alterations in lipid metabolism is not currently known.

Although numerous studies have been conducted on the AKH/AKHR signaling pathway, miRNA-mediated regulation of the AKH signaling pathway at the post-transcriptional level has been little studied. MicroRNAs (miRNAs) are small, non-coding RNAs containing approximately 22 nucleotides that act mostly on the 3' untranslated region (UTR) of target mRNAs, resulting in translation repression or mRNA degradation (*Bartel, 2018*). miRNAs play important roles in regulating cellular events at the post-transcriptional level during biological processes (*Bartel, 2018*; *Asgari, 2011*; *Asgari, 2014*). For example, in *D. citri-C*Las interaction, *C*Las hijacks the JH signaling pathway and host miR-275 that targets the *vitellogenin receptor* (*DcVgR*) to improve *D. citri* fecundity, while simultaneously increasing the replication of *C*Las itself, suggesting a mutualistic interaction in *D. citri* ovaries with *C*Las (*Nian et al., 2024*). In term of host-virus interactions, miR-8, and miR-429 target *Broad isoform Z2*, the gene involved in the hormonal regulation of single nucleopolyhedrovirus (HaSNPV)-mediated climbing behavior in *Helicoverpa armigera* (*Zhang et al., 2018*). In relation to reproduction, in *Aedes aegypti*, miR-275 is essential for egg development (*Bryant et al., 2010*), miR-309 for ovarian development (*Zhang et al., 2016*), and miR-8 for other reproductive processes (*Song and Zhou, 2020*). As for lipid metabolism, knockdown of miR-277 in *A. aegypti* activates insulin/FOXO signaling by increasing the expression of ilp7 and ilp8, which reduces TAG levels and inhibits ovarian development (*Ling et al., 2017*). Nevertheless, the functions of the miRNAs in insect-pathogen interactions need further investigation. To date, there are no reports that miRNAs participate in *D. citri-C*Las interactions or are hijacked by *C*Las to affect lipid metabolism and the fecundity of *D. citri*. As a consequence, in this study, we used the *D. citri-C*Las system as a model to study the molecular events associated with AKH/AKHR signaling that affect lipid metabolism and increase fecundity of *D. citri* induced by *C*Las. Our study contributes to our understanding of the interactions between vectors and pathogens and may also provide new ways for controlling *D. citri* and HLB.

## Results

### Effects of *C*Las infection on the reproduction and metabolism of adult *D. citri*

Total TAG and glycogen levels in fat bodies of *C*Las-negative and *C*Las-positive psyllids were determined. After infection with *C*Las, the TAG and glycogen levels of *C*Las-positive psyllids significantly increased compared with *C*Las-negative individuals (*Figure 1A–B*). Since TAG is mainly stored in fat bodies as lipid droplets, we next evaluated the changes in the lipid droplet size using Nile red staining. As shown in *Figure 1C*, the size of lipid droplets in the *C*Las-positive group was significantly larger than those of the *C*Las-negative group. In order to evaluate the effects of *C*Las on the reproduction of *D. citri*, the preoviposition period, oviposition period, and fecundity of both *C*Las-negative and *C*Las-positive psyllids were examined. After infection with *C*Las, the preoviposition period was significantly shortened, the oviposition period was extended, and the fecundity was markedly increased (*Figure 1D–F*).

### *DcAKH* is involved in changes in metabolism and fecundity in *D. citri* mediated by *C*Las

Using our transcriptome database, cDNA sequences of *DcAKH* were identified and cloned. The cDNA sequence of *DcAKH* is 225 bp in length, encodes a deduced polypeptide of 74 amino acid residues, and has the conserved structural characteristics of a neuropeptide. The conserved structure was 'signal peptide + mature peptide + related peptide,' and the mature peptide sequence of *Dc*AKH is 'QVNFSPNW' (*Figure 2—figure supplement 1A*). A phylogenetic analysis was conducted to evaluate the association of *Dc*AKH with other insect AKHs; *Dc*AKH was most closely related to the AKHs of three hemipteran species, *Nilaparvata lugens*, *Sitobion avenae*, and *Acyrthosiphon pisum* (*Figure 2—figure supplement 1B*).

Tissue expression profiles showed that *DcAKH* was expressed at the highest level in the brain, significantly higher than in the ovaries, fat bodies, and midgut in *D. citri* (*Figure 2—figure supplement 2*; *Figure 2—figure supplement 3*). The levels of *DcAKH* in ovaries of *C*Las-negative and -positive psyllids decreased during the assessment period. However, levels in the ovaries of *C*Las-positive psyllids were significantly higher than those that were *C*Las-negative at all assessment times (*Figure 2A*). To investigate the role of *DcAKH* in metabolic and reproductive changes induced by *C*Las, we knocked

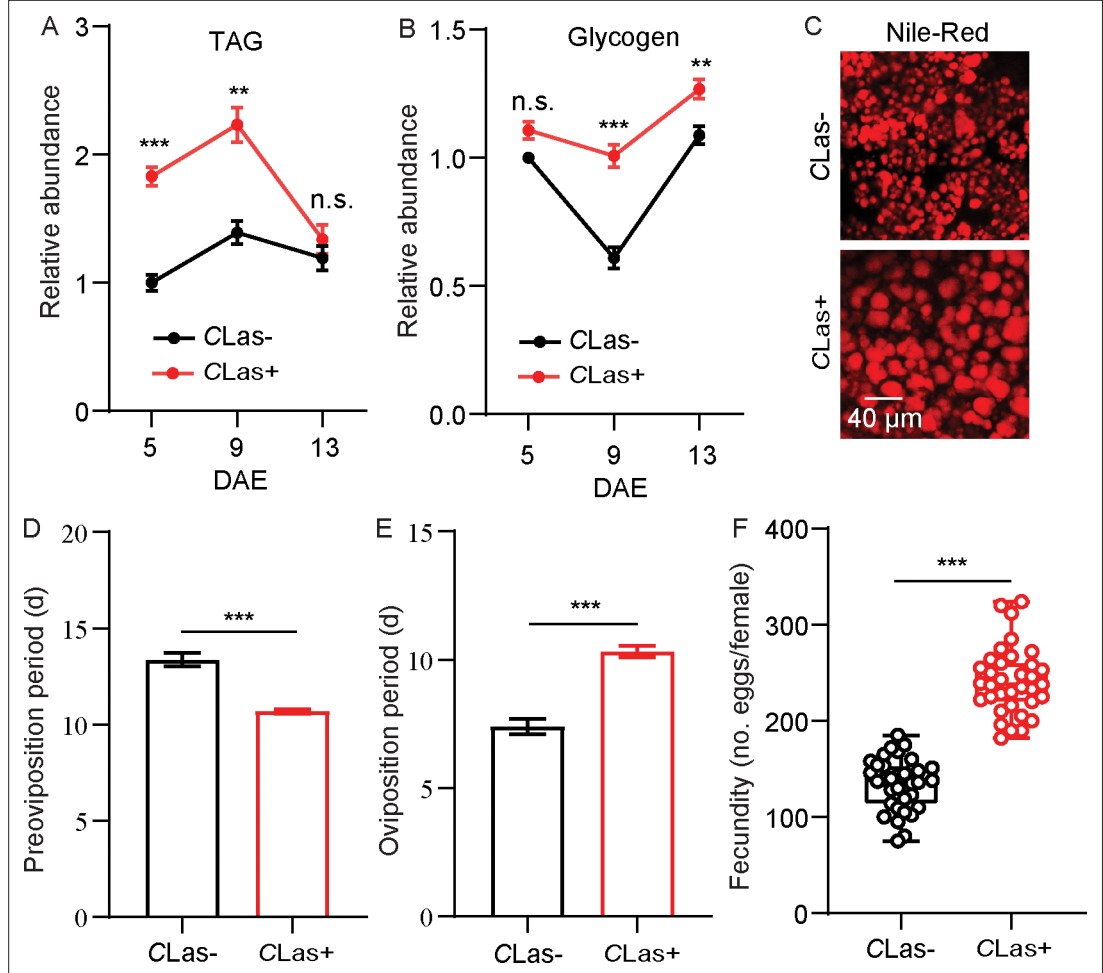

**Figure 1.** Effects of *Candidatus* Liberibacter asiaticus (*C*Las) on lipid metabolism and reproductive behavior of *D. citri*. (**A**) Comparison of triacylglycerides (TAG) levels in the fat bodies of *C*Las-positive (*C*Las+) and *C*Las-negative (*C*Las-) females 5, 9, and 13 days after emergence (DAE). (**B**) Comparison of glycogen levels in the fat bodies of *C*Las + and *C*Las- females 5, 9, 13 DAE. (**C**) Lipid droplets in fat bodies dissected from *C*Las + and *C*Las- females 9 DAE stained with Nile red. Scale bar = 40 µm. (**D–F**) Comparison of the preoviposition period, oviposition period, and the fecundity of *C*Las + and *C*Las- adults. In **A** and **B**, data are shown as means ± SEM with nine biological replications of at least 30 nymphs for each replication. For **D-F**, at least 30 female adults for each group. The significant differences between *C*Las-positive and *C*Las-negative psyllids were determined using Student's *t*-tests (\*\*p<0.01, \*\*\*p<0.001).

down *DcAKH* expression using RNAi. When fed with ds*DcAKH*, the mRNA level of *DcAKH* in *C*Las-negative and *C*Las-positive ovaries significantly decreased by 89% and 66%, respectively, compared with psyllids fed with ds*GFP* (**Figure 2B**). Knockdown of *DcAKH* resulted in TAG accumulation and a significant decrease in glycogen (**Figure 2C–D**). Correspondingly, depletion of *DcAKH* led to a significant increase in the size of the lipid droplets (**Figure 2E**). Compared with the controls, *DcAKH* RNAi slowed ovarian development (**Figure 2F**), extended the preoviposition period (from 13.3 to 14.2 days in *C*Las-negative psyllids and from 10.7 to 13.4 days in *C*Las-positive psyllids), shortened the oviposition period (from 7.1 to 4.3 days in *C*Las-negative psyllids and from 10.1 to 5.7 days in *C*Las-positive psyllids), and decreased fecundity (from 138.0 to 94.7 days in *C*Las-negative psyllids and from 230.0 to 118.1 days in *C*Las-positive psyllids) (**Figure 2G–I**). Furthermore, the *C*Las signal and relative titer in the ovaries of *DcAKH* knockdown psyllids significantly decreased (**Figure 2J–K**). Taken together, it can be concluded that *DcAKH* is involved in *C*Las-induced metabolic and reproductive changes in *D.citri*.

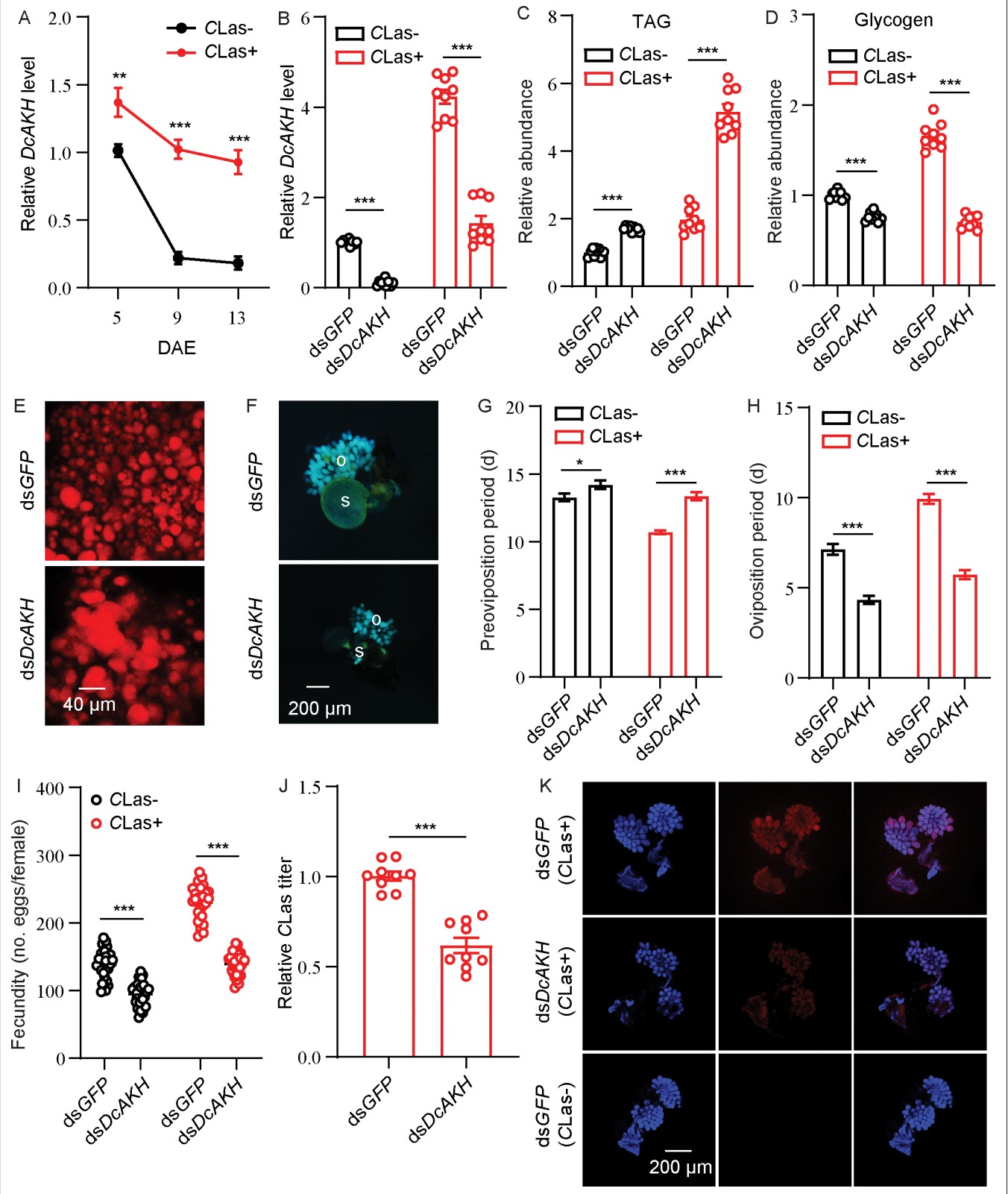

**Figure 2.** *DcAKH* is involved in the mutualistic relationship between *Candidatus* Liberibacter asiaticus (*CLas*) and *D. citri* resulting in increased fecundity. (**A**) Comparison of temporal expression patterns of *DcAKH* between the ovaries of *CLas*- and *CLas*+ psyllids. (**B**) Efficiency of RNAi of *DcAKH* in *CLas*- and *CLas*+ females treated with ds*DcAKH* for 48 hr. (**C**) Comparison of triacylglycerides (TAG) levels in fat bodies of *CLas*- and *CLas*+ females treated with ds*DcAKH* for 48 hr. (**D**) Comparison of glycogen levels in fat bodies of *CLas*- and *CLas*+ females treated with ds*DcAKH* for 48 hr. (**E**) Lipid droplets

*Figure 2 continued on next page*

*Figure 2 continued*

stained with Nile red in fat bodies dissected from *CLas+* females treated with ds*DcAKH* for 48 hr. Scale bar = 40 μm. (**F**) Ovary phenotypes of *CLas+* females treated with ds*DcAKH* for 48 hr. Scale bar = 200 μm. o: ovary, s: spermathecae. (**G–I**) Comparison of the preoviposition period, oviposition period, and the fecundity between *CLas*- and *CLas+* adults treated with ds*DcAKH*. (**J**) The *CLas* titer in ovaries of *CLas+* females treated with ds*DcAKH* for 48 hr. (**K**) Representative confocal images of *CLas* in the reproductive system of *CLas+* females treated with ds*DcAKH* for 48 hr. Scale bar = 200 μm. DAPI: the cell nuclei were stained with DAPI and visualized in blue. *CLas*-Cy3: the *CLas* signal is visualized in red by staining with Cy3. Merge: merged imaging of co-localization of cell nuclei and *CLas*. Data are shown as means ± SEM with at least nine independent biological replications. The significant differences between treatment and controls are indicated by asterisks (Student's *t*-test, *p<0.05, **p<0.01, ***p<0.001).

The online version of this article includes the following figure supplement(s) for figure 2:

**Figure supplement 1.** Sequence characterization of adipokinetic hormone (AKH) from different insect species.

**Figure supplement 2.** Tissue expression of *DcAKH* in *Candidatus* Liberibacter asiaticus (*CLas*)-positive adult females 9 days after emergence in the head, ovary, fat body, and midgut.

**Figure supplement 3.** Melting curve for qRT-PCR primers of *DcAKH*, *DcAKHR*, *Dcβ-ACT*, miR-34, and U6.

## *DcAKHR* responds to *DcAKH* and participates in *D. citri*-*CLas* mutualism

The protein tertiary structure of *DcAKHR* was predicted; *DcAKHR* has seven conserved transmembranes (TM1-TM7) domains and is a typical G protein-coupled receptor (***Figure 3—figure supplement 1A***). A phylogenetic analysis was conducted to evaluate the association of *DcAKHR* with other insect *AKHRs*; *DcAKHR* was most closely related to the *AKHR* of *Nilaparvata lugens* (***Figure 3—figure supplement 1B***). To further confirm *Dc*AKH can activate *DcAKHR*, we performed a cell-based calcium mobilization assay. As shown in ***Figure 3—figure supplement 1C***, *DcAKHR* was strongly activated by *Dc*AKH in a dose-dependent manner, with an $EC_{50}$ value of $2.20 \times 10^{-5}$ mM, but there were no responses when challenged with an empty vector or other evolutionarily-related peptides. In addition, after *DcAKH* knockdown, *DcAKHR* expression in the ovaries significantly decreased (***Figure 3—figure supplement 1D***). *DcAKHR* was cloned, sequenced, designated, and then used for further studies. Sequence analysis showed that *DcAKHR* consisted of an open reading frame (ORF) of 1290 bp and a 185 bp 3' UTR. It encodes a putative protein consisting of 429 amino acids with a predicted molecular weight of 48.32 kDa. Tissue expression patterns showed that *DcAKHR* had the highest expression in the midgut, followed by the head, fat bodies, and ovaries (***Figure 3—figure supplement 2***). In the ovaries of *CLas*-positive and -negative psyllids, the transcript and protein levels of *DcAKHR* increased over the assessment period with levels of both being higher in *CLas*-positive psyllids than in *CLas*-negative psyllids (***Figure 3A***). To investigate the role of *DcAKHR* in metabolic and reproductive changes induced by *CLas*, we knocked down *DcAKHR* expression by RNAi. When fed with ds*DcAKHR*, the mRNA levels of *DcAKHR* in the ovaries of *CLas*-negative and *CLas*-positive were significantly decreased by 59.4 and 66.0%, respectively, compared with psyllids fed with ds*GFP*; western blot analysis confirmed that *Dc*AKHR protein levels were also significantly reduced (***Figure 3B***). Knockdown of *DcAKHR* resulted in TAG accumulation and a significant decrease in glycogen (***Figure 3C–D***). Correspondingly, depletion of *DcAKHR* led to a significant increase in the size of lipid droplets (***Figure 3E***). Compared with the controls, *DcAKHR* RNAi interfered with ovarian development (***Figure 3F***): the preoviposition period was extended (from 13.3 to 16.2 days in *CLas*-negative psyllids and from 10.7 to 12.9 days in *CLas*-positive psyllids); oviposition period was shortened (from 7.1 to 4.6 days in *CLas*-negative psyllids and from 10.1 to 5.3 days in *CLas*-positive psyllids); and fecundity decreased (from 138.0 to 90.6 eggs per female in *CLas*-negative psyllids and from 230.0 to 98.2 eggs per female in *CLas*-positive psyllids). These effects are similar to those caused by *DcAKH* knockdown (***Figure 3G–I***). Furthermore, the *CLas* signal and relative titer in the ovaries of *DcAKH* knockdown psyllids were significantly decreased (***Figure 3J–K***). Taken together, *DcAKHR* responds to *DcAKH* and is involved in the *CLas*-induced metabolic and reproductive changes in *D. citri*.

## miR-34 targeted *DcAKHR*

Based on the small RNA libraries of *D. citri*, we used miRanda and RNAhybrid software to predict the potential miRNAs targeting *DcAKHR*; three putative miRNAs, namely miR-34, miR-2 and miR-14 (***Figure 4A***), were identified. To confirm the binding activity of these miRNAs with *DcAKHR*, several dual-luciferase reporter assays were conducted. Compared with the controls, the luciferase activity

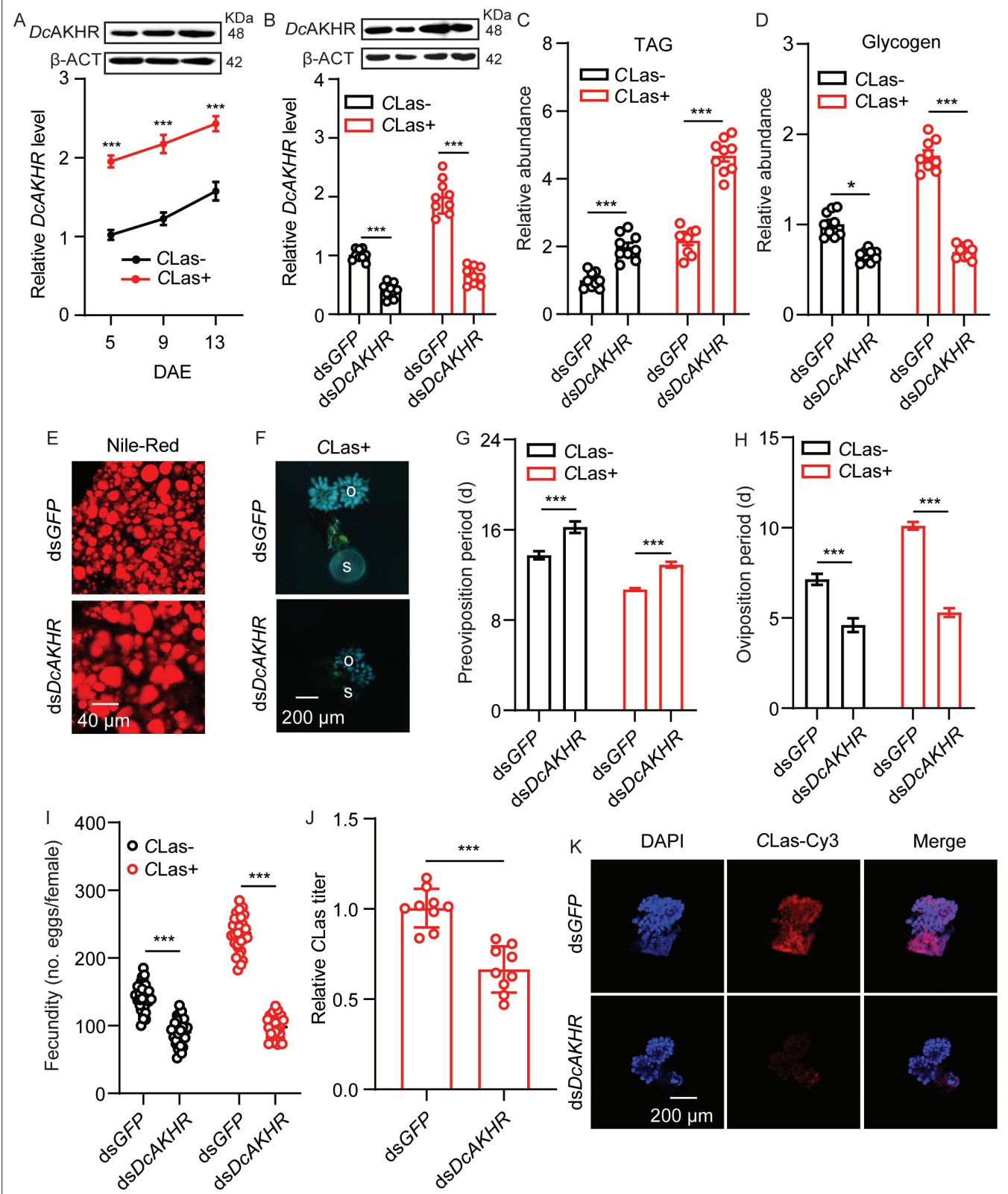

**Figure 3.** *DcAKHR* is involved in the mutualistic relationship between *Candidatus* Liberibacter asiaticus (CLas) and *D. citri* resulting in increased fecundity. (**A**) Comparison of temporal expression patterns of *DcAKHR* between the ovaries of CLas- and CLas+ psyllids. (**B**) The efficiency of RNAi of *DcAKHR* in CLas- and CLas+ psyllids treated with ds*DcAKHR* for 48 h. The protein molecular weight for DcAKHR is 48 KDa and for Dcβ-ACT is 42 KDa. (**C**) Comparison of triacylglycerides (TAG) levels in fat bodies of CLas- and CLas+ females treated with ds*DcAKHR* for 48 hr. (**D**) Comparison of glycogen

*Figure 3 continued on next page*

*Figure 3 continued*

levels in fat bodies of *C*Las- and *C*Las+ females treated with ds*DcAKHR* for 48 hr. (**E**) Lipid droplets stained with Nile red in fat bodies dissected from *C*Las-positive females treated with ds*DcAKHR* for 48 hr. Scale bar = 40 µm. (**F**) Ovary phenotypes of *C*Las+ females treated with ds*DcAKHR* for 48 hr. Scale bar = 200 µm. o: ovary, s: spermathecae. (**G–I**) Comparison of the preoviposition period, oviposition period, and the fecundity of *C*Las- and *C*Las+ adults treated with ds*DcAKHR*. (**J**) The *C*Las titer in the ovaries of *C*Las+ females treated with ds*DcAKHR* for 48 hr. (**K**) Representative confocal images of the reproductive system of *C*Las+ females treated with ds*DcAKHR* for 48 hr. Scale bar = 200 µm. DAPI: the cell nuclei were stained with DAPI and visualized in blue. *C*Las-Cy3: the *C*Las signal is visualized in red by staining with Cy3. Merge: merged imaging of co-localization of cell nuclei and *C*Las. Data are shown as means ± SEM with at least nine independent biological replications. The significant differences between treatment and controls are indicated by asterisks (Student's *t*-test, ***p<0.001).

The online version of this article includes the following source data and figure supplement(s) for figure 3:

**Source data 1.** Original file for the Western blot analysis in *Figure 3A and B* (anti-DcAKHR and anti-β-ACT).

**Figure supplement 1.** In vivo and in vitro studies validating the DcAKH-DcAKHR interaction.

**Figure supplement 2.** Tissue expression of *DcAKHR* in *Candidatus* Liberibacter asiaticus (*C*Las)-positive female adults 9 days after emergence (DAE) in the head, ovary, fat body, and midgut.

was markedly reduced when miR-34 agomir was co-transfected with the recombinant plasmid containing the full 3'UTR sequence of *DcAKHR*, while miR-2 and miR-14 activities were not significantly changed (*Figure 4B*). In addition, the reporter activity was recovered when the binding sites of miR-34 were mutated in the *DcAKHR* 3'-UTR (*Figure 4C*). To determine whether miR-34 specifically targets *DcAKHR*, the following experiments were conducted. First. tissue expression patterns showed that the miR-34 was highly expressed in the head, followed by the fat bodies, ovaries, and midgut (*Figure 4D*). Second, miR-34 has the opposite expression pattern to *DcAKHR* (*Figure 3A*), and transcript levels decreased over the assessment period with the levels of both being lower in *C*Las-positive psyllids than in *C*Las-negative psyllids (*Figure 3—figure supplement 2*). Third, after feeding with agomir-34, the relative levels of miR-34 mRNA in the ovaries of *C*Las-negative and *C*Las-positive psyllids increased 1.9- and 2.3-fold, respectively (*Figure 4—figure supplement 1*). The mRNA and protein levels of *DcAKHR* were correspondingly increased or decreased after separately feeding psyllids with either antagomir-34 or agomir-34 (*Figure 4F*). Fourth, an RNA immunoprecipitation (RIP) assay was performed; *DcAKHR* mRNA was significantly enriched in the anti-AGO-immunoprecipitated RNAs from ovaries of agomir-34-fed female psyllids relative to the control samples (*Figure 4G*). Taken together, these results strongly suggest that *DcAKHR* is a direct target of miR-34.

## miR34 participates in *D. citri*-CLas mutualism in ovaries

To investigate the roles of miR-34 in the interaction between *D. citri* and *C*Las, *C*Las-positive and -negative females were fed with agomir-negative control (NC) or with miR-34 agomir. After feeding with miR-34 agomir, there was a significant accumulation of TAG and a decrease in glycogen levels in the two populations (*Figure 5A–B*). The size of lipid droplets in the fat bodies of *C*Las-positive psyllids were bigger than those of the control (*Figure 5C*). After treatment with miR-34 agomir, ovarian development of *C*Las-positive psyllids was reduced (*Figure 5D*). In addition, when *C*Las-negative and *C*Las-positive psyllids were fed with miR-34 agomir, the oviposition period of both psyllid types was shortened, preoviposition period was markedly extended, and fecundity significantly decreased compared with the relevant control groups (*Figure 5E–G*); these phenotypes mimic those expressed after treatment with ds*DcAKHR*. Moreover, the *C*Las signals and relative titer in *C*Las-positive ovaries were significantly reduced (*Figure 5H–I*). All the results indicate that miR-34 suppresses *DcAKHR* expression and is involved in the mutualistic interaction.

## The JH signaling pathway is regulated by the AKH pathway and is involved in the increase in fecundity induced by CLas

To evaluate whether the JH signaling pathway is regulated by the AKH signaling pathway, we assayed the relative JH titers and the expression levels of several JH pathway-related genes in *DcAKH*-deficient, *C*Las-positive females. Compared to control females, the relative JH titer in the abdomen decreased significantly after ds*DcAKH* treatment for 48 hr (*Figure 6A*). After feeding with ds*DcAKH* for 48 hr, the expression levels of the JH receptor gene, *DcMet*, and the downstream transcription factor, *DcKr-h1*, of the JH signaling pathway were significantly reduced in the fat bodies and ovaries (*Figure 6B–C*). In

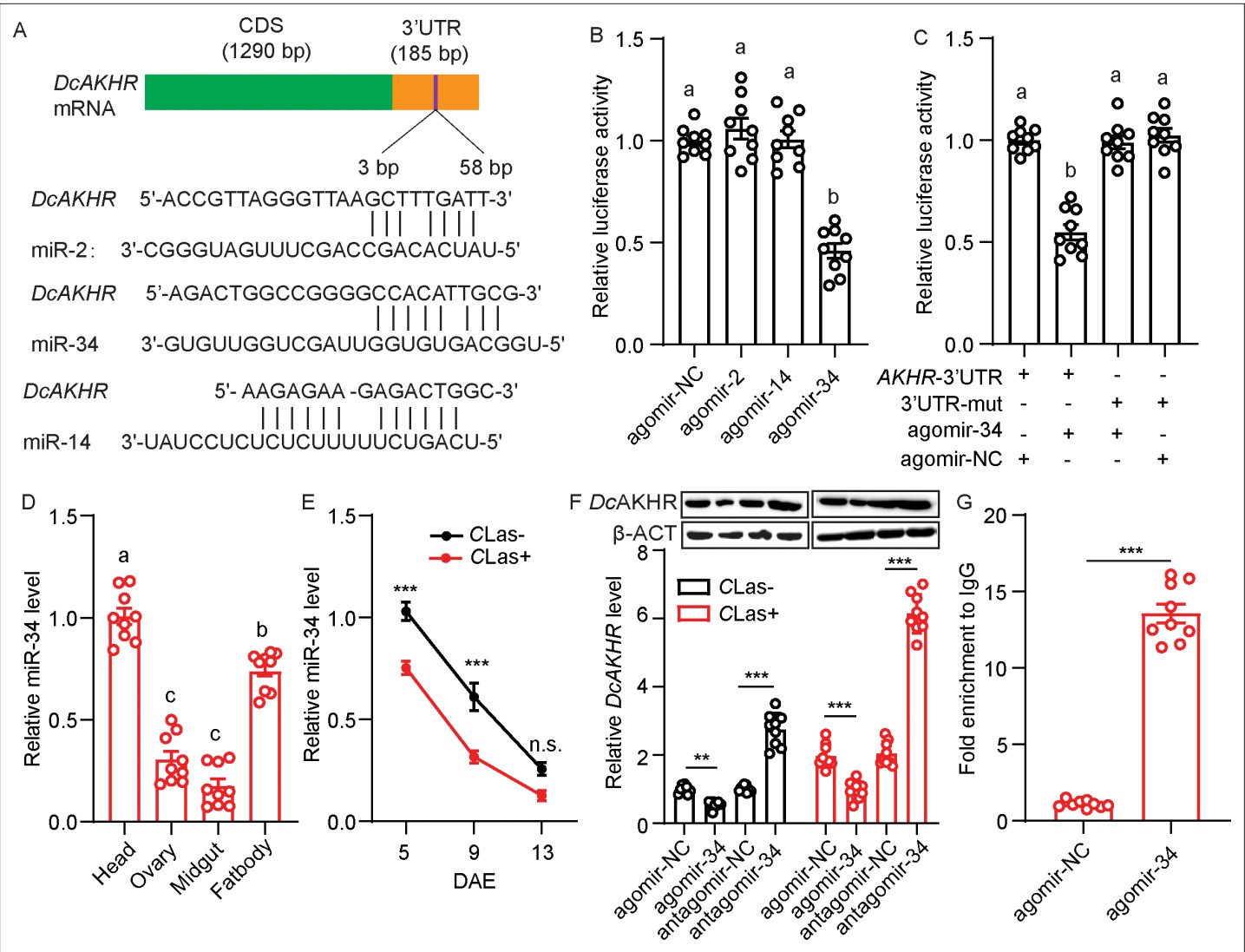

**Figure 4.** Identification and validation of the target relationship between miR-34 and *DcAKHR*. (**A**) The putative binding sites of miRNAs in the *DcAKHR* 3'-untranslated region (UTR) as predicted by miRanda and RNAhybrid. (**B**) Dual-luciferase reporter assays using HEK293T cells co-transfected with miRNA agomir and recombinant pmirGLO vectors containing the predicted binding sites for miR-2, miR-14, and miR-34 in the CDS of *DcAKHR*. (**C**) Dual-luciferase reporter assays using HEK293T cells co-transfected with miR-34 agomir plus recombinant pmirGLO vectors containing *DcAKHR*-3'UTR or mutated *DcAKHR*-3'UTR. (**D**) Tissue expression pattern of miR-34 in *Candidatus* Liberibacter asiaticus (*C*Las) + female adults at 7 days after emergence (DAE) in the head, ovary, fat body, and midgut. (**E**) Comparison of temporal expression patterns of miR-34 in ovaries of *C*Las- and *C*Las + females. (**F**) Effects of miR-34 agomir and antagomir treatments on *Dc*AKHR mRNA expression and protein level in ovaries of *C*Las- and *C*Las + psyllids after 48 hr. The protein molecular weight for DcAKHR is 48 KDa and for Dcβ-ACT is 42 KDa. (**G**) Relative expression of miR-34 targeted *DcAKHR* in vivo as demonstrated by an RNA immunoprecipitation assay. Data are shown as mean ± SEM with nine independent biological replications. For B-D, significant differences among the different treatments are indicated by lowercase letters above the bars (one-way ANOVA followed by Tukey's Honestly Significant Difference test at *p<0.05). The significant differences between treatment and control are indicated by asterisks in E-G (Student's *t*-test, **p<0.01, ***p<0.001).

The online version of this article includes the following source data and figure supplement(s) for figure 4:

**Source data 1.** Original file for the Western blot analysis in *Figure 4F* (anti-DcAKHR and anti-β-ACT).

**Figure supplement 1.** The expression levels of miR-34 in Candidatus Liberibacter asiaticus (*C*Las)-negative and *C*Las-positive females treated with agomir-34 for 48 hr.

addition, the expression levels of *DcVg1-like*, *DcVgA1-like*, and *DcVg*R, three important downstream genes of the JH signaling pathway related to ovarian development were lower in *DcAKH*-deficient *C*Las-positive females than in controls after dsRNA feeding for 48 hr (*Figure 6B–C*). Similar results were observed in the *DcAKHR*-deficient *C*Las-positive females (*Figure 6D–F*) as well as *C*Las-positive

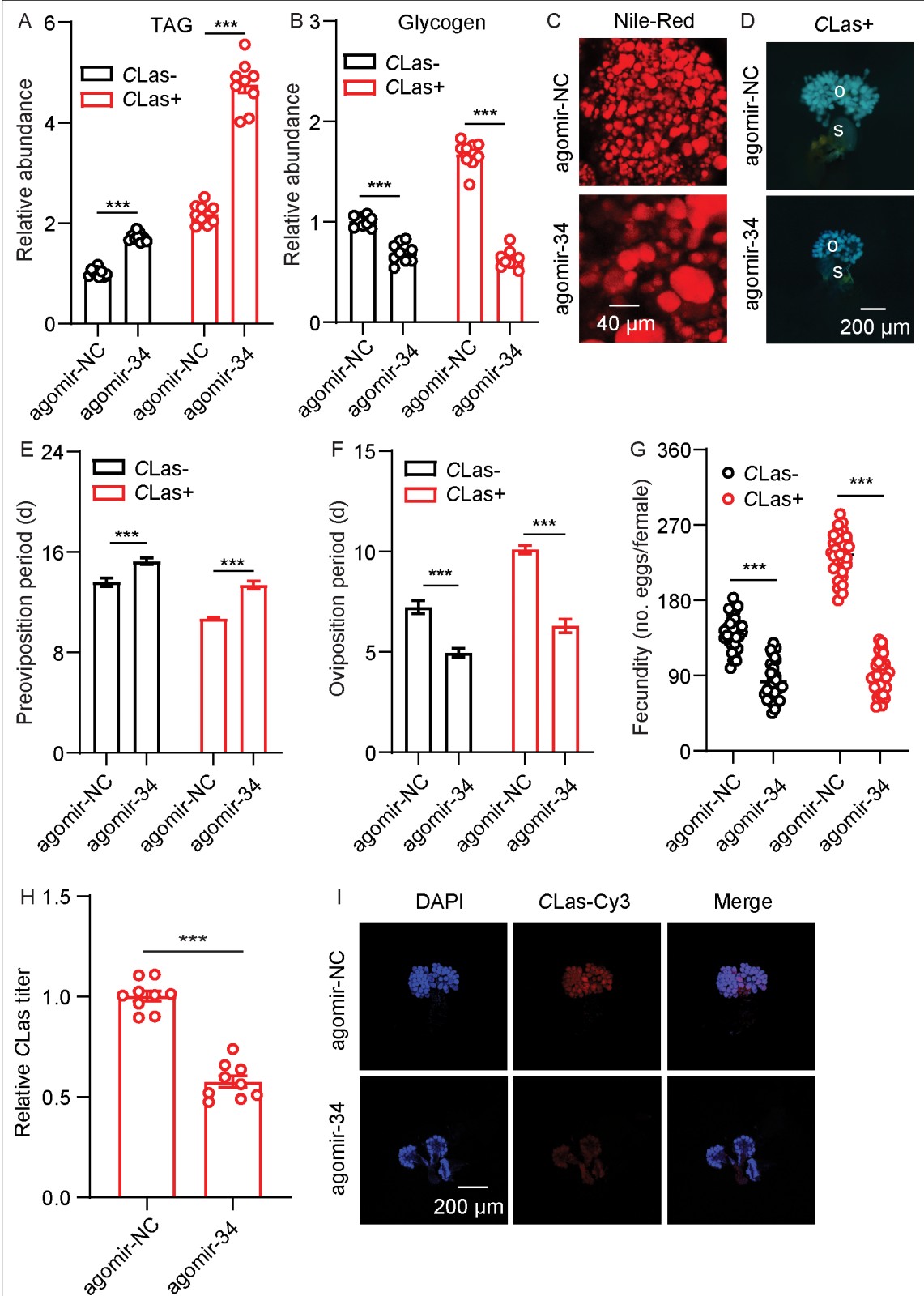

**Figure 5.** miR-34 participation in mutualistic interactions between *D. citri* and *Candidatus* Liberibacter asiaticus (*C*Las). (**A**) Comparison of triacylglycerides (TAG) levels in fat bodies of *C*Las- and *C*Las + females treated with agomir-34 for 48 hr. (**B**) Comparison of glycogen levels in the fat bodies of *C*Las- and *C*Las + females treated with agomir-34 for 48 hr. (**C**) Lipid droplets stained with Nile red in fat bodies dissected from *C*Las + females treated with agomir-34 for 48 hr. Scale bar = 40 μm. (**D**) Ovary phenotypes of *C*Las + female treated with agomir-34 for 48 hr. Scale bar =

*Figure 5 continued on next page*

*Figure 5 continued*

200 µm. o: ovary, s: spermathecae. (**E–G**) Comparison of the preoviposition period, oviposition period, and the fecundity between *C*Las- and *C*Las + adults treated with agomir-34. (**H**) *C*Las titer in ovaries of *C*Las + females treated with agomir-34 for 48 hr. (**I**) Representative confocal images of *C*Las in the reproductive system of *C*Las +females treated with agomir-34 for 48 hr. Scale bar = 200 µm. The signals of DAPI and *C*Las-Cy3 are the same as described in *Figure 2*. Data are shown as means ± SEM with at least nine independent biological replications. The significant differences between treatment and controls are indicated by asterisks (Student's *t*-test, ***$p < 0.001$).

females fed with agomir-34 (*Figure 6G–I*). These results suggest that JH signaling pathway is regulated by the AKH-AKHR-miR-34 signaling pathway and is involved in the increased fecundity of *D. citri* induced by *C*Las.

## Discussion

An increasing number of studies have focused on the effects of vector-virus interactions on reproduction. For example, rice gall dwarf virus is transmitted vertically by hitchhiking on insect sperm, thereby promoting long-term viral epidemics and persistence in nature (*Mao et al., 2019*). *Berasategui et al., 2022* found a mutualistic relationship between *Chelymorpha alternans* and *Fusarium oxysporum*, where the pupae were protected in exchange for the dissemination and propagation of the fungus. Although there is limited research on the mechanisms underlying vector-bacteria interactions. In *D. citri-C*Las interaction, *C*Las operates host hormone signaling and miRNA to mediate the mutualistic interaction between *D. citri* fecundity and its replication (*Nian et al., 2024*). *Singh and Linksvayer, 2020* found that *Wolbachia*-infected colonies of *Monomorium pharaonis* exhibited increased colony-level growth, accelerated colony reproduction, and shortened colony life cycles compared to uninfected colonies.

The fat body is the major insect storage organ that produces the majority of hemolymph-born vitellogenic proteins and critically contributes to insect ovary development (*Arrese and Soulages, 2010*). In this study, after infection with *C*Las, the preoviposition period of *D. citri* was shortened, oviposition period was prolonged, and fecundity was significantly increased. These results suggest that there is a mutualistic interaction in *D. citri* ovaries with *C*Las. *C*Las accelerates ovarian development and prolongs the oviposition period thereby enhancing the fecundity of *D. citri*; in turn, *D. citri* provides sites and nutrients for *C*Las replication during ovarian development. *C*Las-positive *D. citri* requires more energy to support this increased fecundity. Hence, the TAG and glycogen levels of both *C*Las-positive and *C*Las-negative were assayed. After infection with *C*Las, the TAG and glycogen levels significantly increased, and the average size of lipid droplets in the fat bodies were distinctly larger, indicating that the increase in lipid mobilization may prepare for an increase in fecundity.

The total amount of lipids present in fat bodies is a balance between lipogenesis and lipolysis (*Grönke et al., 2005*), and both processes are essential for lipid homeostasis. Disruption of either lipogenesis or lipolysis could interfere with nutritional and physiological conditions in fat bodies and affect vitellogenesis. AKH/AKHR mediated lipolytic systems are essential for lipid provision and nutrient transfer during reproduction. In *Gryllus bimaculatus* (*Lorenz, 2003*; *Lorenz and Gäde, 2009*) and *Caenorhabditis elegans* (*Lindemans et al., 2009*), injections of AKH interfered with egg production by inhibiting the production of vitellogenins and the accumulation of lipid stores. In *Bactrocera dorsalis*, AKHR knockdown led to decreased lipolytic activity and delayed oocyte maturation (*Hou et al., 2017*) and these authors suggested the decline in fecundity may be due to the inability to utilize the lipid stores of fat bodies to promote oocyte maturation. In *Glossina morsitans*, the silencing of *AKHR* and *brummer* led to the inability to mobilize lipid reserves in fat bodies during pregnancy, slowed down the development of oocytes, and caused embryogenesis to fail, indicating that female fertility depends on lipid metabolism regulated by AKHR (*Attardo et al., 2012*). These results suggest that lipid homeostasis appears critical for insect fecundity, as its disruption substantially suppresses egg production and negatively affects vitellogenesis.

In the study, *DcAKH* and *DcAKHR* played important roles in the increased lipid metabolism and fecundity of *D. citri* mediated by *C*Las. The cDNA sequences encoding AKH and AKHR from *D. citri* were cloned and identified, and phylogenetic analysis showed that *DcAKH* and *DcAKHR* were orthologous to other hemipteran AKHs and AKHRs. Functional analysis showed that *DcAKHR* shares properties found with other AKHRs. *DcAKHR*-transfected CHO cells could be activated by AKH peptide,

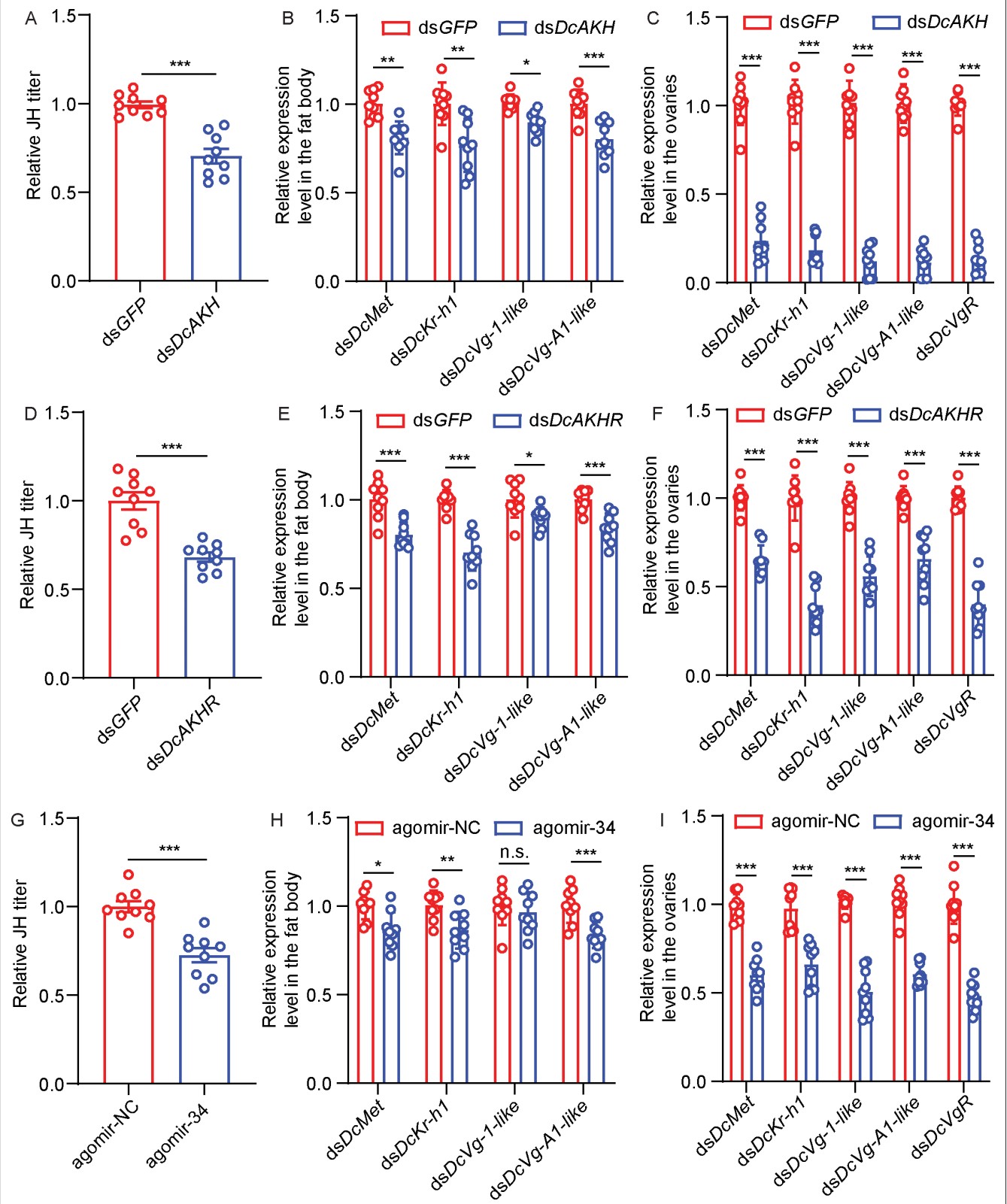

**Figure 6.** The juvenile hormone (JH) signaling pathway is regulated by adipokinetic hormone (AKH) signaling pathway and is involved in the increase in fecundity of *D. citri* induced by *Candidatus* Liberibacter asiaticus (*C*Las). (**A**) JH titer in the abdomen of *C*Las + females treated with ds*DcAKH* for 48 hr. (**B**) Effects of ds*DcAKH* treatment on mRNA level of JH signaling pathway in fat bodies of *C*Las + females. (**C**) Effects of ds*DcAKH* treatment on mRNA levels of components of the JH signaling pathway in the ovaries of *C*Las + females. (**D**) JH titers in the abdomens of *C*las + females treated with

*Figure 6 continued on next page*

*Figure 6 continued*

ds*DcAKHR* for 48 hr. (**E**) Effects of ds*DcAKHR* treatment on mRNA level of JH signaling pathway in fat bodies of *C*Las + females. (**F**) Effects of ds*DcAKHR* treatment on mRNA levels of components of the JH signaling pathway in ovaries of *C*Las + females. (**G**) JH titer in abdomen of *C*Las + females treated with agomir-34 for 48 hr. (**H**) Effects of agomir-34 treatment on mRNA levels of components of the JH signaling pathway in fat bodies of *C*Las + females. (**I**) Effects of agomir-34 treatment on mRNA levels of components of the JH signaling pathway in the ovaries of *C*Las + females. Data are shown as means ± SEM with at least nine independent biological replications.

while it did not respond to Crz peptide, despite the sequences and gene structures of AKH and Crz being closely related (**Park et al., 2002**). *DcAKH* and *DcAKHR* were more highly expressed in the ovaries of *C*Las-positive psyllids than those of *C*Las-negative individuals. Knockdown of *DcAKH* and *DcAKHR* resulted in TAG accumulation and a significant decrease in glycogen levels in the fat bodies. Most importantly, the *C*Las titer in the ovaries of *C*Las-positive psyllids and fecundity significantly reduced when feeding with *dsDcAKH* and *dsDcAKHR*. Our results support the idea that reproductive disruption from AKH/AKHR knockdown is due to the inability of females to mobilize lipid reserves in the fat bodies required for oocyte maturation during vitellogenesis.

To date, there are no reports on the post-transcriptional regulation of AKHR. As important post-transcriptional regulators, miRNAs generally suppress their target genes by triggering mRNA degradation or translational repression (**Bartel, 2018**). There is increasing evidence implicating miRNAs in the metabolic processes of insects, particularly in relation to reproduction. In addition to functions of miR-277 described in the Introduction, this miRNA targets ilp7 and ilp8 thereby controlling lipid metabolism and reproduction in *A. aegypti* (**Ling et al., 2017**). miR-8 depletion in mosquitoes resulted in an increase of its target gene, *secreted wingless-interacting molecule*, and lipid accumulation in developing oocytes (**Lucas et al., 2015**). Depletion of miR-8 in mosquitoes by antagomir injection upregulates *secreted wingless-interacting molecule* and decreases the level of lipids in ovaries to block vitellogenesis post-blood meals (**Page et al., 2012**). A switch from catabolic to anabolic amino acid metabolism in fat bodies mediated by miR-276 restricts mosquito investment into oogenesis and benefits the development of *Plasmodium falciparum* (**Lampe et al., 2019**). In the current study, based on in vitro and in vivo experiments, we found that host miR-34 targets *DcAKHR*. miR-34 was observed to have an opposite expression trend to *DcAKHR*, and its expression in the ovaries of *C*Las-positive psyllids was lower than in those of *C*Las-negative individuals. Treatment of miR-34 agomir and antagomir, respectively, resulted in a significant decrease and increase in *DcAKHR* mRNA and protein levels. In addition, miR-34 overexpression not only led to the accumulation of TAG and the reduction of glycogen levels in fat bodies, but also decreased fecundity and *C*Las titers in the ovaries similar to the phenotypes caused by silencing of *DcAKHR*. This indicates that *C*Las inhibited host miR-34 to enhance *DcAKHR* expression and improve the lipid metabolism and fecundity. This study is the first to report the manipulation of host miRNAs and target genes by a bacterium to affect lipid metabolism and reproduction of their vector.

Vitellogenesis is strongly linked with insect JH production (**Roy et al., 2018**). The fat body is an important tissue that senses and integrates various nutritional and hormonal signals required for the regulation of vitellogenesis (**Arrese and Soulages, 2010**). Moreover, JH is also involved in the nutrient-dependent regulation of insect reproduction, but the mechanism of action remains obscure. **Lu et al., 2018**; **Lu et al., 2016a**; **Lu et al., 2016b** have demonstrated that nutritional signaling during female reproduction induces JH biosynthesis that in turn stimulates vitellogenin expression in fat bodies and egg production in *N. lugens*. In order to evaluate the possible relationship between AKH/AKHR-mediated lipolysis and JH-dependent vitellogenesis, the expressions of JH signaling pathway-related genes in fat bodies and ovaries were determined after knockdown of *DcAKH* and *DcAKHR*. We found that reducing *DcAKH* or *DcAKHR* by RNAi in *C*Las-positive *D. citri* not only decreased JH titer but also resulted in a decrease in the expression of *DcMet* and *DcKr-h1* in the fat bodies and ovaries, as well as the egg development-related genes, *DcVg-1-like*, *DcVg-A1-like* and *DcVgR*. Similar results were detected in the agomir-34 treatment. Our results suggest that AKH/AKHR-based lipolysis plays an important regulatory role in the JH signaling pathway during *D. citri-C*Las interactions.

In conclusion, we proposed the scheme of events following *C*Las infection presented in *Figure 7*. Upon infection with *C*Las, *D. citri* exhibits enhanced fecundity compared to uninfected individuals. The interaction between *C*Las and *D. citri* affecting reproduction is a win-win strategy; the increased offspring of *D. citri* contributes to a higher presence of *C*Las in the field. *C*Las upregulates the AKH/

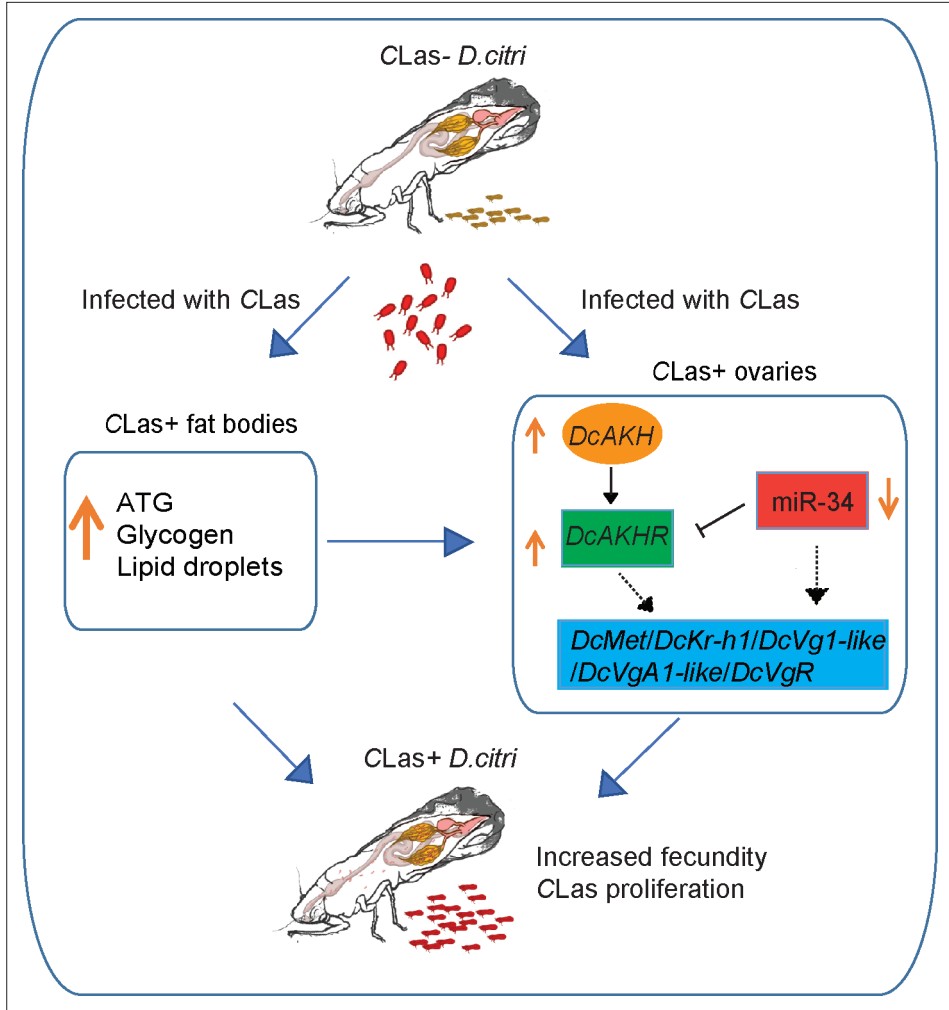

**Figure 7.** Mechanisms linking metabolism and reproduction of *D. citri* induced by *Candidatus* Liberibacter asiaticus (*C*Las). After infection with *C*Las, the triacylglycerides (TAG) and glycogen levels in fat bodies of *C*Las-positive psyllids significantly increased as well as the size of lipid droplets. In ovaries, *C*Las upregulates the AKH/AKHR signaling and downregulates miR-34 to increase lipid metabolism and activate juvenile hormone (JH)-dependent vitellogenesis, thereby improving the fecundity of *C*Las-positive females. *D. citri* are more fecund than their uninfected counterparts. The interaction between *C*Las and *D. citri* affecting reproduction is a win-win strategy; the more offspring of *D. citri*, the more *C*Las in the field.

AKHR signaling and downregulates miR-34 to increase lipid metabolism and activate JH-dependent vitellogenesis, thereby improving the fecundity of *C*Las-positive females. Overall, our results not only broaden our understanding of the molecular mechanism underlying the *D. citri-C*Las interaction, but also provide valuable insights for further integrated management of *D. citri* and HLB by targeting the *DcAKH/DcAKHR* signaling pathway.

## Materials and methods
### Host plants and insect colonies
The *C*Las-negative (healthy) and *C*Las-positive *D. citri* colonies were obtained from a laboratory culture continuously reared on the healthy and *C*Las-infected lemon (*Citrus ×limon* (L.) Osbeck) plants. Monthly monitoring of the *C*Las infection in both the lemon plants and psyllids was conducted using quantitative polymerase chain reaction (qPCR). The two populations were raised in different incubators with same conditions (26±1°C, 65±5% relative humidity (RH), and a 14 hr light: 10 hr dark cycle). *C*Las-negative and *C*Las-positive lemon plants were grown in two separate glasshouses.

## Bioinformatics and phylogenetic analyses

The physicochemical properties of *DcAKH* (GenBank accession number: MG550150.1) and *DcAKHR* (GenBank accession number: OR259432) were analyzed using the online bioinformatics ProtParam tool (http://web.expasy.org/protparam/). Search and download amino acid sequences of other species in the NCBI database. Homologous protein sequences of AKH and AKHR were aligned with those of other insects using DNAman6.0.3 software. Phylogenetic trees were constructed with the neighbor-joining method in MEGA5.10 software, with 1000 bootstrap replicates; bootstrap values >50% are shown on the tree.

## Assays of reproductive parameters

Tested females from two colonies were selected and paired with healthy males. One female and one male were placed onto young vegetative shoots (flush) of healthy lemon plants to promote oviposition; the flush and insects were sealed with a tied white mesh bag (15×20 cm). These lemon plants were held in an incubator (25±1°C, 65±5% RH, 14 L: 10D photoperiod). After 24 hr, the preoviposition period and oviposition period were recorded, the number of eggs laid per female were counted, and this pair of psyllids were transferred to a new flush to continue egg laying. Eggs were counted daily until all females died. All the experiments were performed using three replications, and about 15 pairs of *D. citri* for each replication.

## Quantitative RT-PCR (qRT-PCR)

For mRNA expression analysis, total RNA was extracted using TRIzol reagent (Invitrogen, Carlsbad, CA, United States), and the synthesis of first-strand cDNA was performed using a PrimeScript II 1st Strand cDNA Synthesis Kit (Takara, Beijing, China) in accordance with the manufacturer's instructions. qRT-PCR was carried out by using the TB Green Premix Ex Taq II (Takara, Beijing, China) on an ABI PRISM 7500 Real-Time System (Applied Biosystems, Foster City, CA, USA). The beta-actin (*Dcβ-ACT*, GenBank XM_026823249.1) gene was used as an internal control to normalize gene expression levels. For miRNA expression analysis, miRNA was extracted using a miRcute miRNA Isolation Kit (TIANGEN, Beijing, China), synthesized using a miRcute Plus miRNA First-Strand cDNA Kit (TIANGEN, Beijing, China), and quantified using the miRcute Plus miRNA qPCR Kit (SYBR Green) (TIANGEN, Beijing, China). U6 snRNA was used as an internal control to normalize the expression of miRNA. All the primers used are listed in *Supplementary file 1*.

## Luciferase activity assay

The 185 bp sequence of the 3'UTR containing the predicted target sites for miR-34 in *DcAKHR* was cloned into the pmirGLO vector (Promega, Wisconsin, USA) downstream of the luciferase gene to construct the recombinant plasmid, *DcAKHR*-3'UTR-pmirGLO, using the pEASY-Basic Seamless Cloning and Assembly Kit. The mutated 3'UTR sequence was amplified and cloned into the pmirGLO vector to generate the *DcAKHR*-3'UTR mutant-pmirGLO plasmid. The agomir and antagomir of miRNAs were chemically synthesized and modified by GenePharma (Shanghai, China) with chemically modified RNA oligos of the same sequence or anti-sense oligonucleotides of the miRNA. The negative control for agomir and antagomir was provided by the manufacturer. According to the manufacturer's instructions, the constructed vector (500 ng) and miRNA agomir (275 nM) were co-transferred into HEK293T cells in a 24-well plate using a Calcium Phosphate Cell Transfection Kit (Beyotime, Nanjing, China). After co-transfection for 24 hr, the activities of firefly and Renilla luciferase were detected using the Dual-Glo Luciferase Assay System (Promega, Madison, WI, USA), and the average luciferase activity was calculated.

## Fluorescence in situ hybridization (FISH)

FISH using a *C*Las probe was carried out as previously described (*Hosseinzadeh et al., 2019*). Under a dissecting microscope, the ovaries of *C*Las-negative and *C*Las-positive *D. citri* were dissected in 1×phosphate-buffered saline (PBS) (Beyotime Biotechnology, Shanghai, China). Firstly, the isolated ovaries were fixed in Carnoy's fixative (glacial acetic acid-ethanol-chloroform, 1:3:6, vol/vol) for 12 hr at 25 °C, washed four times (5 min each time) with 6% $H_2O_2$ in 80% ethanol and then three times (10–15 min per time) with PBST (1x PBS: TritonX-100, 99.7:0.3, (vol/vol)). Second, the samples were pre-incubated three times (10 min per time) in hybridization buffer (20 mM Tris-HCl, pH 8.0, 0.9 M

NaCl, 30% formamide (vol/vol), 0.01% sodium dodecyl sulfate (wt/wol)) containing 10 pmol/mL of each probe for 24 hr at 25 °C, then washed three times with PBST. Third, nuclei were stained with 0.1 mg/mL of 40, 60-diamidino-2-phenylindole (DAPI) (Sigma-Aldrich, St Louis, MO, USA) for 15 min, then washed once with PBST. Finally, the stained ovaries were mounted in a mounting medium. The slides were scanned under a Leica TCS-SP8 (Leica Microsystems Exton, PA USA) confocal microscope; excitation lasers emitting at 405 nm and 550 nm were used to detect the DAPI and Cy3 signals, respectively, and sequential scanning was used to avoid the signal overlap of probes. Image processing was completed using Leica LAS-AF software (v2.6.0). Specificity of detection was carried out using without the probe and CLas-negative controls. Three FISH tests were performed and at least 20 ovaries were viewed under the microscope to confirm the repeatability. The probe sequences used in the current study are listed in *Supplementary file 1*.

## Western blotting

The total proteins from *D. citri* ovaries were extracted using RIPA protein lysis buffer (50 mM Tris pH 7.4, 1% Triton X-100, 150 mM NaCl, 1% sodium deoxycholate, and 0.1% SDS) with 1 mM PMSF. Protein concentrations were quantified using a BCA protein assay kit (Beyotime, Jiangsu, China). Aliquots of 60 µg protein were separated by 12% SDS-PAGE and transferred to polyvinylidene fluoride membranes (Millipore). Subsequently, the membranes were incubated with the primary antibody against *DcAKHR* (1: 1000 dilution, ABclonal Technology Co., Ltd., Wuhan, China) for 12 hr at 4 °C and then with the secondary antibody (goat anti-rabbit IgG conjugated with HRP, 1: 10000 dilution) for 2 hr at 25 °C. A mouse monoclonal antibody against β-actin (TransGen Biotech, Beijing, China) was used as a control. Immunoreactivity was imaged with the enhanced chemiluminescence with Azure C600 multifunctional molecular imaging system.

## Nile red staining

The fat bodies were dissected from adult females 5 DAE, fixed in 4% paraformaldehyde for 30 min at 25°C, and washed twice with 1x PBS. Lipid droplets were incubated for 30 min within the mixture of Nile red (0.1 µg/µL, Beijing Coolaber Technology Co., Ltd) and DAPI (0.05 µg/µL), and then washed twice with 1x PBS. The samples were imaged using a laser scanning microscopy (TCS-SP8, Leica Microsystems Exton, PA USA).

## Determination of TAG levels

TAG levels were measured using a Triglycerides Colorimetric Assay (Cayman) in accordance with the manufacturer's instructions. In short, thirty fat bodies were homogenized in 100 µL of Diluent Assay Reagent. Then 10 µL of supernatant was incubated with an Enzyme Mixture solution, and the TAG contents from the measurements were normalized to protein levels in the supernatant of the samples determined using a BCA protein assay (Thermo Fisher Scientific). Three independent biological replicates were analyzed for each treatment, and each treatment included three technical replicates.

## Determination of glycogen levels

Glycogen levels were assayed using a Glycogen Assay Kit (Cayman) in accordance with the manufacturer's instructions. In brief, thirty fat bodies were homogenized in 100 µL of Diluent Assay Reagent, and 10 µL of supernatant was incubated with Enzyme Mixture solution and Developer Mixture. The glycogen levels from the measurements were normalized to protein levels in the supernatant of the sample determined using a BCA protein assay (Thermo Fisher Scientific). Three independent biological replicates were analyzed for each treatment, and each treatment included three technical replicates.

## dsRNA synthesis and RNAi assay

dsRNAs of *DcAKH* and *DcAKHR* were transcribed by a Transcript Aid T7 High Yield kit (Thermo Scientific, Wilmington, DE, United States) and purified using the GeneJET RNA Purification kit (Thermo Scientific). miRNA agomir, agomir negative control, miRNA antagomir, and antagomir negative control were synthesized in the Shanghai GenePharma Co. Ltd. (Shanghai, China). The RNAi and miRNA treatments were performed by feeding dsRNA or miRNA antagomir/agomir through an artificial diet as described previously (*Yu and Killiny, 2018*). Briefly, 20 females at 7 DAE were placed into a glass cylinder (25×75 mm) and sealed with two stretched paramembranes. 200 µL of 20% (w:v) sucrose

mixed with dsRNA was placed between two paramembranes for feeding. The final ingestion concentrations of ds*DcAKH* and ds*DcAKHR* were 200 ng/µL and 300 ng/µL, respectively. After feeding with dsRNA for 48 hr, the treated females were assayed as follows. The ovaries were dissected or qRT-PCR and western blot analysis. Reproductive parameters including the pre-oviposition period, oviposition period, and fecundity were recorded. *C*Las titers in the ovaries were assayed. Ovary morphologies were observed using an Ultra-Depth Three-Dimensional Microscope (VHX-500). Fat bodies were stained with Nile red. TAG levels and glycogen levels were determined. Green fluorescent protein (GFP) was used as a reporter gene. Feeding ds*GFP* was used as the control when psyllids were fed with dsRNA. All the experiments were performed in three replications, and each replication included 15–20 pairs of *D. citri*.

## Heterologous expression and calcium mobilization assay

The complete ORF of *DcAKHR* was cloned into the expression vector pcDNA3.1$^+$; cloning was confirmed by sequencing (TSINGKE Bio). Endotoxin-free plasmid DNA of the vector was extracted, and all tested peptides were synthesized by Sangon Biotech (Shanghai, China) at a purity >95% (see *Figure 1B* for peptide structures). Chinese hamster ovary (CHO-WTA11) cells supplemented with Gα16 subunit and aequorin were used (*Jiang et al., 2017*) and transfected with pcDNA3.1$^+$-*DcAKHR* or empty pcDNA3.1$^+$ (negative control) vectors using Lipofectamine 2000 (Thermo Fisher Scientific, Waltham, MA USA). The calcium mobilization assay was performed with slight modification as described by *Gui et al., 2017* (*Gui et al., 2017*) and *Shi et al., 2017* (*Shi et al., 2017*). In brief, the cells were incubated for 3 hr in the dark with coelenterazine h (Promega, Madison, WI, USA) 48 hr after the transfection. Then, serial dilutions of peptide ligands ($10^{-6}$ to $10^1$ µM) were loaded into opaque 96-well plates. Luminescence was measured over 15 s using a SpectraMax i3x Multi-Mode Microplate Reader (Molecular Devices). Medium alone was used as a blank control and 100 µM ATP was served as a positive control. Each experiment was repeated three times.

## JH titer detection

JH titers were assessed using the Insect JH III ELISA kit (Shanghai Enzyme-linked Biotechnology Co., Ltd. Shanghai, China) following the manufacturer's protocol.

## Statistical analysis

All statistical values are presented as means ± SEM. Statistical analyses were performed by GraphPad Prism 8.0 software. Student's *t*-test was used to determine the pairwise comparisons at the following significance levels (*$p<0.05$, **$p<0.01$, ***$p<0.001$). For multiple comparisons, one-way ANOVA with Tukey's Honest Significant Difference tests was used to separate means at $p<0.05$.

# Acknowledgements

This research was supported by the National Natural Science Foundation of China, grant number 32102193, and the Open Competition Program of Ten Major Directions of Agricultural Science and Technology Innovation for the 14th Five-Year Plan of Guangdong Province, grant number 2022SDZG07.

# Additional information

### Funding

| Funder | Grant reference number | Author |
| --- | --- | --- |
| National Natural Science Foundation of China | 32102193 | Xiaoge Nian |

| Funder | Grant reference number | Author |
|---|---|---|
| The Open Competition Program of Ten Major Directions of Agricultural Science and Technology Innovation for the 14th Five-year Plan of Guangdong Province | 2022SDZG07 | Yurong He |

The funders had no role in study design, data collection and interpretation, or the decision to submit the work for publication.

## Author contributions

Jiayun Li, Conceptualization, Resources, Data curation, Software, Formal analysis, Validation, Investigation, Visualization, Methodology, Writing – original draft; Paul Holford, George Andrew Charles Beattie, Conceptualization, Validation, Visualization, Methodology, Writing – review and editing; Shujie Wu, Jielan He, Resources, Data curation, Investigation; Shijian Tan, Investigation; Desen Wang, Software, Formal analysis; Yurong He, Conceptualization, Funding acquisition, Visualization, Methodology; Yijing Cen, Conceptualization, Resources, Funding acquisition, Visualization, Methodology, Writing – original draft, Project administration, Writing – review and editing; Xiaoge Nian, Conceptualization, Resources, Data curation, Software, Formal analysis, Funding acquisition, Investigation, Visualization, Methodology, Writing – original draft, Project administration, Writing – review and editing

## Author ORCIDs

Xiaoge Nian (iD) http://orcid.org/0000-0001-6341-6431

Reviewer #1 (Public Review): https://doi.org/10.7554/eLife.93450.3.sa1
Reviewer #2 (Public Review): https://doi.org/10.7554/eLife.93450.3.sa2
Author response https://doi.org/10.7554/eLife.93450.3.sa3

# Additional files

## Supplementary files

• Supplementary file 1. Primer list used for 3'RACE, qRT-PCR, and RNAi analysis. Related to Method details.

## Data availability

The published article includes all data generated or analyzed during this study. The full sequences were submitted to NCBI, accession number: MG550150.1 for DcAKH, OR259432 for DcAKHR, OP251123 for DcMet, XM_026820026.1 for DcKr-h1, XM_008488883.3 for DcVg-1-like, XM_026832896.1 for DcVg-A1-like and OP251122 for DcVgR.

The following datasets were generated:

| Author(s) | Year | Dataset title | Dataset URL | Database and Identifier |
|---|---|---|---|---|
| Nian X | 2024 | Diaphorina citri adipokinetic hormone receptor protein mRNA, complete cds | https://www.ncbi.nlm.nih.gov/nuccore/OR259432 | NCBI GenBank, OR259432 |
| Nian X | 2022 | Diaphorina citri juvenile hormone receptor methoprene-tolerant protein (Met) mRNA, complete cds | https://www.ncbi.nlm.nih.gov/nuccore/OP251123 | NCBI GenBank, OP251123 |
| Nian X | 2022 | Diaphorina citri vitellogenin receptor (VgR) mRNA, complete cds | https://www.ncbi.nlm.nih.gov/nuccore/OP251122 | NCBI GenBank, OP251122 |

The following previously published datasets were used:

| Author(s) | Year | Dataset title | Dataset URL | Database and Identifier |
|---|---|---|---|---|
| Wang Z, Zeng X | 2017 | Diaphorina citri AKH mRNA, complete cds | https://www.ncbi.nlm.nih.gov/nuccore/MG550150.1/ | NCBI GenBank, MG550150.1 |
| Saha S | 2013 | PREDICTED: Diaphorina citri Krueppel homolog 1-like (LOC103524696), mRNA | https://www.ncbi.nlm.nih.gov/nuccore/XM_026820026.1/ | NCBI GenBank, XM_026820026.1 |
| Saha S | 2017 | PREDICTED: Diaphorina citri vitellogenin-1-like (LOC103523873), mRNA | https://www.ncbi.nlm.nih.gov/nuccore/XM_008488883.3 | NCBI GenBank, XM_008488883.3 |
| Saha S | 2013 | PREDICTED: Diaphorina citri vitellogenin-A1-like (LOC103523199), mRNA | https://www.ncbi.nlm.nih.gov/nuccore/XM_026832896.1 | NCBI GenBank, XM_026832896.1 |

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
