## [Editor Report · eLife assessment]

This **important** study reveals the molecular basis of mutualism between a vector insect and a bacterium responsible for the most devastating disease in citrus agriculture worldwide. The evidence supporting the conclusions is **compelling**, with **solid** biochemical and gene expression analyses demonstrating the phenomenon. We believe this work will be of great interest to the fields of vector-borne disease control and host-pathogen interaction.

---

## [Referee Report · Reviewer #1 (Public Review)]

Summary:

The manuscript by Jiayun Li and colleagues aims to provide insight into adipokinetic hormone signaling that mediates the fecundity of Diaphorina citri infected by 'Candidatus Liberibacter asiaticus'. CLas-positive D. citri are more fecund than their CLas-negative counterparts and require extra energy expenditure. Using FISH, qRT-PCR, WB, RNAi, and miRNA-related methods, authors found that knockdown of DcAKH and DcAKHR not only resulted in triacylglycerol accumulation and a decline of glycogen but also significantly decreased fecundity and CLas titer in ovaries. miR-34 suppresses DcAKHR expression by binding to its 3' untranslated region, whilst overexpression of miR-34 resulted in a decline of DcAKHR expression and CLas titer in ovaries and caused defects that mimicked DcAKHR knockdown phenotypes.

---

## [Referee Report · Reviewer #2 (Public Review)]

Diaphorina citri is the primary vector of Candidatus Liberibacter asiaticus (CLas), but the mechanism of how D. citri maintains a balance between lipid metabolism and increased fecundity after infection with CLas remains unknown. In their study, Li et al. presented convincing methodology and data to demonstrate that CLas exploits AKH/AKHR-miR-34-JH signaling to enhance D. citri lipid metabolism and fecundity, while simultaneously promoting CLas replication. These findings are both novel and valuable, not only have theoretical implications for expanding our understanding of the interaction between insect vectors and pathogenic microorganisms but also provide new targets for controlling D. citri and HLB in practical implications. The conclusions of this paper are well supported by data.

---

## [Author Response]

The following is the authors’ response to the original reviews.

Key Considerations:There seem to be two inconsistencies related to some results depicted in Figures 1, 2, 3 and 5.Firstly, Figure 1 shows the effect on _C_Las infection (_C_Las+) compared to the control (_C_Las-), where results show an increase of TAG, Glycogen, lipid droplet size, oviposition period, and fecundity. In Figures 2, 3, and 5, the authors establish the involvement of the genes *DcAKH*, *DcAKHR*, and miR34 in this process, by showing that by preventing the function of these three factors the effects of _C_Las+ are lost. However, while Figure 1 shows the increase of TAG and lipid droplet size in _C_Las+, Figures 2, 3, and 5 do not show a significant elevation in TAG when comparing _C_Las- and _C_Las+.Secondly, in addition to the absence of statistical difference in TAG and lipid droplet size observed in Figure 1, Figures 2, 3, and 5 show an increase in TAG and lipid droplet size after ds_DcAKH_ (Figure 2), ds_DcAKHR_ (Figure 3) and agomiR34 (Figure 5) treatments. Considering that AKH, AKHR, and miR34 are important factors to _C_Las-induce increase in TAG and lipid droplet size, one might expect a reduction in TAG and lipid droplet size when _C_Las+ insects are silenced for these factors, contrary to the observed results.

Thanks for your excellent suggestion. Lipid droplets are cellular organelles responsible for storing lipids within cells, playing a crucial role in fat metabolism and energy homeostasis. The formation and breakdown of lipid droplets involve a complex interplay of genes and enzymes, including DGAT (for synthesis), ATGL and HSL (for breakdown). In _C_Las-negative *D. citri*, there is a delicate balance between creasing and breaking down of lipid droplets. The enlargement of lipid droplet size following _C_Las infection may result from a significantly higher synthesis rate compared to breakdown, as more energy is required during early ovarian development. The hormone AKH, a key player in fat metabolism, primarily stimulates fat breakdown. Therefore, when *DcAKH* and *DcAKHR* are silenced without affecting fat synthesis, there is no enhancement of fat breakdown; instead, there is an accumulation of lipid droplets, resulting in their enlargement. This suggests that _C_Las infection affects both the breakdown and synthesis of lipid droplets, while AKH and AKHR primarily impact the breakdown, leading to similar outcomes. However, the underlying physiological mechanisms warrant further in-depth exploration.

**Reviewer #1 (Recommendations For The Authors):**
(1) Line 25: change "In addition" to "Additionally".

Thanks for your wonderful suggestion. We have changed “In addition” to “Additionally” in our revised manuscript (Line 26).

(2) Lines 60-72: Have there been any previous reports on the interaction between host AKH hormones and microorganisms in insects or animals? If yes, please add more background.

Thanks for your wonderful suggestion. We have added the interactions between host AKH hormones and microorganisms in insects (Line 74-81).

(3) Lines 82-95: add the following reference about the miR-275 of *Diaphorina citri* in the background. Nian, X., Luo, Y., He, X., Wu, S., Li, J., Wang, D., Holford, P., Beattie, G. A. C., Cen, Y., Zhang, S., & He, Y. (2024). Infection with '*Candidatus* Liberibacter asiaticus' improves the fecundity of *Diaphorina citri* aiding its proliferation: A win-win strategy. Molecular Ecology, 33, e17214.

Thanks for your wonderful suggestion. We have added the sentence “in *D. citri*-_C_Las interaction, _C_Las hijacks the JH signaling pathway and host miR-275 that targets the *vitellogenin receptor* (*DcVgR*) to improve *D. citri* fecundity, while simultaneously increasing the replication of _C_Las itself, suggesting a mutualistic interaction in *D. citri* ovaries with _C_Las” in our revised manuscript (Line 97-100).

(4) In the figures of Nile red staining, the digit of the scale bar should be added.

Thanks for your wonderful suggestion. We have added the digit of the scale bar for Nile red staining in the Figure 1C, 2E, 3E, 5C.

(5) In Figures 2G-H, 3G-H, 5E-F, the presentation of data should be consistent with Figure 1D-E.

Thanks for your wonderful suggestion. We have changed figure 1D-E in our revised manuscript.

(6) In the discussion part, more information should be added about miR-275 and *DcVgR* from the above reference.

Thanks for your wonderful suggestion. We have added the information “In *D. citri*-_C_Las interaction, _C_Las operates host hormone signaling and miRNA to mediate the mutualistic interaction between *D. citri* fecundity and its replication” in Line 350-353.

(7) For the primer specific, please add the melting curves for qPCR primers of *DcAKH*, *DcAKHR*, *Dcβ-ACT*, U6, and miR-34 in the supplementary material.

Thanks for your wonderful suggestion. We have added the melting curves for qPCR primers of *DcAKH*, *DcAKHR*, *Dcβ-ACT*, U6 and miR-34 in the supplementary material of Figure S6.

(8) Line 476: *Dcβ-ACT* was indicated as a gene and should be Italic.

Thanks for your wonderful suggestion. We have changed “DcβACT” to “*Dcβ-ACT*” in our revised manuscript (Line 491).

(9) Reference style should be consistent and correct. Like [5], [10], [37], [47].

Thanks for your wonderful suggestion. We have revised them in our revised manuscript.

**Reviewer #2 (Recommendations For The Authors):**
(1) In order to better engage readers, I suggest emphasizing the "enhanced fecundity" in the title. A suggestion for the revised title is: Adipokinetic hormone signaling mediates the enhanced fecundity of *Diaphorina citri* infected by '*Candidatus* Liberibacter asiaticus'.

Thanks for your wonderful suggestion. We have changed the title to “Adipokinetic hormone signaling mediates the enhanced fecundity of *Diaphorina citri* infected by '*Candidatus* Liberibacter asiaticus'” in our revised manuscript.

(2) For the abstract, in lines 14-15, please change the first sentence to "*Diaphorina citri* serves as the primary vector for '*Candidatus* Liberibacter asiaticus' (_C_Las), the bacterium associated with the severe Asian form of huanglongbing." In line 18, delete "present". In line 19, change "increased" to "increasing". In line 21, change "triacylglycerol accumulation" to "the accumulation of triacylglycerol". In line 33, change "in *D. citri* ovaries with _C_Las" to "between _C_Las and *D. citri* ovaries".

Thanks for your wonderful suggestion. We have revised them following your suggestion in our revised manuscript, including changed “*Diaphorina citri* is the primary vector of the bacterium, ‘*Candidatus* Liberibacter asiaticus’ (_C_Las) associated with the severe Asian form of huanglongbing” to “*Diaphorina citri* serves as the primary vector for '*Candidatus* Liberibacter asiaticus' (_C_Las), the bacterium associated with the severe Asian form of huanglongbing” in Line 15-16; deleted "present" in Line 19; changed "increased" to "increasing" in Line 20; changed "triacylglycerol accumulation" to "the accumulation of triacylglycerol" in Line 22; changed "in *D. citri* ovaries with _C_Las" to "between _C_Las and *D. citri* ovaries" in Line 34.

(3) In lines 57-59, change "How *D. citri* maintains a balance between lipid metabolism and increased fecundity after infection with _C_Las is not known." to "However, the mechanism of how *D. citri* maintains a balance between lipid metabolism and increased fecundity after infection with _C_Las remains unknown.".

Thanks for your wonderful suggestion. We have changed " How *D. citri* maintains a balance between lipid metabolism and increased fecundity after infection with _C_Las is not known" to "However, the mechanism of how *D. citri* maintains a balance between lipid metabolism and increased fecundity after infection with _C_Las remains unknown" in our revised manuscript (Line 58-60).

(4) In Figure 1, "n.s" should be changed to "n.s.", "n.s." should be added in 13 DAE of Figure 1A, and the specific numerical value of the scale bar should be indicated on Figures 1C, 2E, 3E, and 5C.

Thanks for your wonderful suggestion. We have revised them in our revised manuscript.

(5) In all the figure legends, the "**P < 0.01,***P < 0.001" should be changed to "**p < 0.01,***p < 0.001".

Thanks for your wonderful suggestion. We have revised them in our revised manuscript.

(6) In Figures 1D-E, the preoviposition period and oviposition period were presented using a box diagram, but in other figures (including Figure 2G-H, Figure 3G-H, Figure 5E-F) these were shown using a column chart. Please keep the method of presentation consistent.

Thanks for your wonderful suggestion. We have revised the figure 1D-E in our revised manuscript.

(7) For discussion, in line 333, change "Increasing numbers" to "An increasing number". In line 334, change "vertically transmitted" to "transmitted vertically".

Thanks for your wonderful suggestion. We have changed "Increasing numbers" to "An increasing number" in Line 345; changed "vertically transmitted" to "transmitted vertically" in Line 346 in our revised manuscript.

(8) In lines 338-342, change "There are few studies on the mechanisms underlying vector-bacteria interactions. However, Singh and Linksvayer (2020) [38] found that *Wolbachia*-infected colonies of *Monomorium pharaonis* had increased colony-level growth, accelerated colony reproduction, and shortened colony life cycles compared to those that were uninfected." to "Although there is limited research on the mechanisms underlying vectorbacteria interactions, Singh and Linksvayer (2020) [38] found that _Wolbachia_infected colonies of *Monomorium pharaonis* exhibited increased colony-level growth, accelerated colony reproduction, and shortened colony life cycles compared to uninfected colonies.".

Thanks for your wonderful suggestion. We have revised it in our revised manuscript (Line 350-355) .

(9) In line 370, delete "present". In lines 386-387, change "More and more miRNAs have been reported to be involved in the metabolic processes of insects including reproduction." to "There is increasing evidence implicating miRNAs in the metabolic processes of insects, particularly in relation to reproduction.".

Thanks for your wonderful suggestion. We have revised them in our revised manuscript, including deleted "present" in Line 383 and changed "More and more miRNAs have been reported to be involved in the metabolic processes of insects including reproduction" to "There is increasing evidence implicating miRNAs in the metabolic processes of insects, particularly in relation to reproduction" in Line 399-400.

(10) In line 423, change "After infection with _C_Las, *D. citri* are more fecund than their uninfected counterparts." to "Upon infection with _C_Las, *D. citri* exhibits enhanced fecundity compared to uninfected individuals.". In lines 424-425 and 439-440, change "the more offspring of *D. citri*, the more _C_Las in the field" to "the increased offspring of *D. citri* contributes to a higher presence of _C_Las in the field.". In Line 429, change " information" to "insights".

Thanks for your wonderful suggestion. We have revised them in our revised manuscript, including changed "After infection with _C_Las, *D. citri* are more fecund than their uninfected counterparts" to "Upon infection with _C_Las, *D. citri* exhibits enhanced fecundity compared to uninfected individuals" in Line 436-437; changed "the more offspring of *D. citri*, the more _C_Las in the field" to "the increased offspring of *D. citri* contributes to a higher presence of _C_Las in the field" in Line 438-439; changed "information" to "insights" in Line 443.

(11) In lines 446-447, change "The _C_Las-infected lemon plants and psyllids were monitored to detect _C_Las infection monthly using the quantitative polymerase chain reaction (qPCR)" to "Monthly monitoring of the _C_Las infection in both the lemon plants and psyllids was conducted using quantitative polymerase chain reaction (qPCR)".

Thanks for your wonderful suggestion. We have revised it in our revised manuscript (Line 460-461).

(12) In lines 452-458, how did the authors identify homologous sequences of AKH and AKHR for phylogenetic tree analysis and alignment of the amino acid sequences? From NCBI or other databases? The methodological details should be added.

Thanks for your wonderful suggestion. We have added the methodological details in our revised manuscript (Line 469-470).

(13) In line 476, Dcβ-ACT should be italic.

Thanks for your wonderful suggestion. We have changed “DcβACT” to italic in our revised manuscript (Line 491).

(14) In line 538, the manufacturer should be provided for Nile Red.

Thanks for your wonderful suggestion. We have provided the manufacturer of Nile Red in our revised manuscript (Line 553).

(15) Does miR-34 have any other target genes? If yes, whether they have any function in the fecundity improvement of D. citri after infected by CLas.

Thanks for your insightful suggestion. In addition to *DcAKHR*, we predicted three other genes have binding sites in 3’UTR with miR-34, including Innexin, T-box transcription factor TBX1, and fatty acid synthase. Despite this, the mRNA expression levels of all three genes remained unchanged between _C_Las-negative and _C_Las-postive females. Therefore, we believe that these genes are not implicated in the fecundity improvement.

(16) The reference format should be unified. Please revise references 10, 28, 43, 47, and 53.

Thanks for your wonderful suggestion. We have revised them in the revised manuscript.